# New Business Models Based on Multiple Value Creation for the Customer: A Case Study in the Chemical Industry

**Iveta Šimberová** *[ID] **and Peter Kita**

Institute of Management, Faculty of Business and Management, Brno University of Technology, Kolejni 2906/4, 61200 Brno, Czech Republic; kitap@fbm.vutbr.cz
* Correspondence: simberova@fbm.vutbr.cz

**Abstract:** The paper's objective is to describe business models currently used in terms of sustainable multiple customer creation in the chemical industry in the Czech Republic, namely Section 20.1 in the CZ NACE (Classification of Economic Activities). The business models are described through a specified set of business model elements, which correspond with the presented theoretical bodies. The business models are also evaluated and benchmarked based on a custom indicator measuring business model novelty. The theoretical background of the research consists of three theoretical bodies: Sustainability, multiple customer value creation, and new business models. The research stems from the theoretical background and anticipates that the business model development dynamics drives companies to consider the reasons and conditions of their very existence. The Canvas business model serves as a visualization tool, as it is sufficiently comprehensive, analytical, flexible, and general. For this reason, it is appropriate for the research of new business models aimed at multiple value creation in any industry. Owing to the frequency of occurrence of elements in the fields of the canvas business model, it is possible to develop the majority and minority business model design representing the basis of the research.

**Keywords:** sustainable value creation; chemical industry; multiple value; majority business models; minority business models

## 1. Introduction

The current globalisation of markets and constant new social challenges demanding the preservation and strengthening of the competitiveness of corporate groups promote the search for new ways of creating the multiple value for customers. Innovative efforts aimed at creating the new value for customers can be qualified as a new area of innovation that leads to the creation of new business models. The framework within which this value is created is linked to sustainability. New business models within the perspective of sustainability take in synchronisation of a number of values (economic, social, and environmental) and are anchored together with all aspects of the multiple value. Feedback from customers is the foundation for the modification of business models and their further refinement. The issue of new business models and the creation of multiple value for the customer is an extensive and interdisciplinary one. Both these concepts represent new approaches to the interpretation of the concepts of the business model and value, and our analysis of available expert sources has revealed that the issue of new business models, with a focus on the sustainable creation of multiple value for the customer, is a developing area. This is substantiated by a growing number of papers and various research studies on the given topic. The basial issue for further research is the frequency of occurrence of elements in the fields of the canvas business model, and possibility to develop the majority and minority business model design [1].

*The Context of the Investigated Issue*

The dynamics of socioeconomic relationships, including changes in the behaviour of individual market entities, result in new forms of the approach to concepts and methods of managing contemporary concerns in the area of managerial scientific disciplines. In Europe in the second half of the 20th century, these changes relate, first and foremost, to the investigation of the relationships between environmental issues and the effects of the economic system: A shortage of raw materials, the problems of waste management, global warming, and pollution of the air, soil, water, etc. These problems led to the creation of a new context favouring sustainable development in which new models of production and consumption are formed and new business models are developed with the aim of creating multiple value for the customer. The report on trends in sustainable development (Sustainability and Reporting Trends in 2025–Preparing for the Future) drawn up by the international organisation Global Reporting Initiative [2] emphasises the responsibility of corporate groups in the area of sustainable development [2]. In addition to environmental issues, sustainable development also takes in the sustainability of economic growth and social cohesion [3]. The social responsibility of corporate groups represents the voluntary incorporation of social and ecological interests into their business strategy and into the daily activities of these corporate groups and their relationships with stakeholders [4].

We selected the chemical industry, which is a growing branch of industry both in the Czech Republic [5] and in Europe as a whole, as a suitable area for our research. Globalisation is posing an ever-increasing threat to the position of the European and Czech chemical industry. In order to preserve its competitive position on foreign markets, it should develop activities in two directions [6]:

- Restoring margins by assuring access to energy at competitive prices and adopting measures prioritising investment promoting productivity;
- Assuring the transition to sustainable chemicals by means of innovative endeavours (the transition to other sources of energy and the development of plant chemistry represent the principal factors in development).

Corporate groups in the chemical industry in the Czech Republic in the group 20.1 according to the CZ NACE classification were selected for the purposes of this research.

An assessment of the current state of knowledge shows that it is essential for these corporate groups to assure the attainment of the goal of economic efficiency in line with social justice and protection of the environment [7]. The given challenges are, at the same time, an incentive for the inception of new requirements of business modelling. The aim of our research project was to respond to the current challenges of sustainable development, in the form of new business models for selected groups of the Czech chemical industry 20.1 according to the CZ NACE classification.

## 2. Theoretical Background

### 2.1. Sustainable Innovations

Innovations may, in general terms, take various forms, e.g., innovation of a product or process, organisational, technological, or business innovation, or changes to the business model. According to [8], innovations are a mechanism for the creation of value and a vehicle for competitiveness. They demand creativity, a vision, prediction of the future, and entrepreneurial experience. All the given aspects aid the creation of new business models [9]. The principle of permanent [10] or intensive innovations follows a hyperactive and mass-operative model. It is characterised by the frequent replacement and purchase of products. For corporate groups, this means producing more products more quickly with the aim of greater consumption [11]. Nevertheless, this intensive model of consumption has wide-ranging consequences for the environment, as it is accompanied by an increase in the amount of waste and the wastage of limited natural resources and raw materials. This creates a potential tension between the economic and environmental dimensions of the intensive industrial model. The integration of sustainable development into innovative strategies creates the principal benefit for the corporate group.

In this regard, [12] puts forward the concept of sustainable innovation of the business model. This relates to the discovery of a new and improved method for the functioning of the corporate group and the creation of greater value [13]. Sustainable innovations of the business model are considered crucial to the creation of sustainable business [14]. Sustainable innovations respond to the challenges of sustainable development and integrate its environmental, social, and economic dimensions into the attributes and functionalities of products and, most importantly, enable the development of new business models in dependence on the branch of industry in which the corporate group conducts its activity and in which it must also respect the legislative aspects of its activity. Sustainable innovations contribute in this way to shaping the reputation and legitimacy of the corporate group [15], i.e., they demonstrate that its activities do not have a negative impact on the environment and society. Their development is associated with the corporate group's stakeholders who are a source of information. According to [16], value is created by corporate groups which work together with internal and external stakeholders within the scope of the corporate group's value network. The aim of sustainable innovation in the context of sustainable development is the creation of new market opportunities that lead to the development of new business models focusing on multiple value for customers.

*2.2. The Concept of A New Business Model*

A new business model is, then, an innovative business model [17,18] as it relates to a new system of company activities [19] and an innovative structure of creating and obtaining value [20] for the corporate group and its alliances with partners and customers [21]. Amit and Zott [22] understand networks and alliances as a key framework for innovative business models. Teece [23] notes that traditional business models solve the task of how to develop, manufacture, and sell a product and receive profit from its development, manufacture, and sale. Developing a new business model, in the broadest terms, does not need to consist of developing a new product or a change in processes or the sources necessary to its production, and in addition, do not necessarily require to be of a technological nature. The desired effects may also be achieved by means of faster processes, the sale of solutions to customers, the provision of accompanying services, co-operation with communities, price incentives for customers, breaking down barriers to the availability of a product, etc. According to [9], a business model may be considered new if it satisfies two principles: Uniqueness of concept and uniqueness of sales proposition.

2.2.1. Uniqueness of Concept

The uniqueness of concept of a new business model consists of the fact that it:

- Is focused on long-term sustainability and is, according to the authors Bocken, Short, Rana, and Evans [24], a concept for a sustainable business model;
- Is characterised by an environmental, social, and economic dimension [24], which is the result of the environmental awareness of corporate group, which thereby create the possibility of unique ways of doing business. In this respect, it is a concept for a socially responsible model [25];
- Is associated with innovation. In this regard, it is a concept for innovative business models [26–28].

The concept of sustainability plays an important basic role in the formulation of a corporate group's strategic vision and its decision-making [29,30]. It is a strategic turning point for a large number of corporate groups in that it brings purpose to socially responsible enterprise [31] and is associated with a culture of innovation, unceasing improvement, and a set of moral principles and standards that guide behaviour in the sphere of enterprise [32]. The creation of merely economic value cannot be sustainable for a corporate group ignoring the environmental and social setting in which it performs its activity.

A new business model in the context of sustainability is the result of strategic thinking and an operative scheme for improved business [33]. Description of this scheme emphasises the uniqueness of the activity of the corporate group. It considers the relationship between its strategic vision and operative activities in view of the fact that it directly integrates the environmental, social, and economic

aspect of sustainable development into its strategic and operative approach. The creation of multiple value is the basis of this approach.

### 2.2.2. Uniqueness of Sales Proposition

The acquisition of a positive economic effect on the basis of sales is based on the use of a unique selling proposition, which means a description of how a specific product differs from competitor products in terms of its environmental, social, and economic effect in order for the customer to decide to buy it. The products themselves that a corporate group offers may help sustainability to only a limited extent unless they give consideration to a broader set of questions, including the social interests of stakeholders, questions of human rights, and the preservation of sources, etc. [34]. Richardson [35] proposes conceiving a new business model from a broader perspective: Firstly, not merely for customers and corporate groups, but also for all interest groups (suppliers, end users, shareholders, governments, and partners), and secondly, value is related not merely to its economic expression, but also incorporates environmental and social value. The value supply may be considered an expression of the corporate group's social and environmental responsibility.

### 2.3. Multiple Value

The qualification of the concept of multiple value is the result of an understanding of the set of definitions and the value theory that are specific for each area of management [36]. The approach based on value for the customer is of interest, first and foremost, to marketing [37]. Socially responsible products are designed for recipients who are not merely external, but also internal, customers. Vlček [38] emphasises the fact that economic entities are, in their real form, an extensive network of mutually interconnected and interrelated processes with internal and external products creating a source of value for internal and external customers. Value is characterised by multiplicity by the fact that it is associated with environmental and social aspects. This multiplicity, with which the given concept is associated, follows from Bergson's philosophy [39]. It denotes the structure created on the basis of the conjunctive synthesis of individual specific elements. It is a mutual operation: Symbiosis and sympathy. It relates to the participation of stakeholders in the creation of multiple value, which represents a new approach to the value that the corporate group offers. Social and environmental values are in accordance with contemporary requirements of producing and commercialising socially responsible products [40] that respect the collective interests of society. A shift is being seen from the concept of the product to the concept of the supply of solutions on the basis of a corporate group's network interconnection [41] with other stakeholders (suppliers, customers, not-for-profit organisations, etc.). Continual multiplicity enables a better understanding of the nature of value creation, which is a concept that lies at the very heart of marketing [39].

The concept of multiple value results from adaptation of the dimensions of sustainable development (environmental, social, and economic) and the concept of value. It expresses a new corporate approach to value that is open to customer thinking in the context of sustainable development and that incorporates:

- The perceived benefit of a product with a differing period of consumption from the viewpoint of its use, competitive advantages, and the possibilities of differentiation by external and internal customers;
- The innovative dimension associated with the provision of services, the production process. and distribution making faster production and distribution possible at lower cost, while respecting environmental and social aspects;
- The process of co-operation during its creation within the framework of a system of multiple interactions of interest groups in a company value network and the co-creation of unique value with the customer;
- The challenge for the manufacturer of ensuring their development at a profit, while concurrently respecting the costly requirements of sustainable development.

According to the academic literature, the interest of corporate groups is becoming focused on the creation of higher value (extraordinary or new value) for the customer, which we understand as multiple value and which is associated with the environmental, social, and economic aspects of supply. The specific interest is an opportunity for stimulation of the sustainable innovation of not merely products and processes, but of business models as well. It follows from the above that the process of innovation goes beyond the bounds of the product itself. If multiple value is to be created for the customer; it is essential to expand the value of the product by innovating associated services and the process of production and distribution making it possible to make and distribute products at lower cost, at greater quality, and at higher speed, while at the same time observing environmental and social aspects. The creation of multiple value [42] anticipates dialogue with interest groups that reinforce the ability to innovate and defend new sources of multiple value. The alliance of various interest groups within a corporate group evokes their mutual complementation and knowledge sharing with an awareness of the fact that the creation of value for all interest groups is, first and foremost, a matter of creating value for the customer [43].

The particular expert views based on approaches to value emphasise the idea that the resolution of social and environmental issues does not mean a burden on corporate groups but represents an opportunity to improve their business models and increase profitability.

## 2.4. The Architecture of Multiple Value

When we recognise the value that the customer buys and the values that other partners expect, attention must be focused on the method of their creation and delivery.

The value architecture is the way in which the corporate group is organised in order to ensure the creation and provision of value supply to end customers on the basis of a portfolio of sources and competences. It is associated not merely with the internal and external value chain, but first and foremost with its value network, i.e., the corporate group's external stakeholders who help create value and provide it to customers. Its aim within the value chain is to ensure the environmentally and socially effective creation of value.

Co-operation between partners in the value chain and the integration of various processes in marketing, distribution, innovation, information systems, etc. have become important factors in achieving customer satisfaction, reducing total costs, and creating the competitive position of various entities in the value chain [44].

The advantages of a sustainability strategy, which are incorporated primarily in a progressive approach (Table 1) [45–47], should be part of the creation of a value chain [48].

**Table 1.** A differing approach to the value chain from the viewpoint of creation of economic, environmental, and social value.

| The Traditional Approach | The Progressive Approach |
|---|---|
| • The structure of the corporate process is set by the organisation created | • Intensive co-operation between the corporate group's various departments |
| • The division of functions and responsibilities is crucial, corresponding to the organisational structure | • Co-operation between corporate groups and other stakeholders creates environmental, social and economic value |
| • Planned relationships are comprised exclusively of relationships between the corporate group, its suppliers and its customers | • The multi-organisational aspect enables the incorporation of creation of multiple value into the enterprise ecosystem |
| • The corporate group's primary activities do not consider the value of natural resources or the value of waste resulting from consumption | • Co-operation between partners is an important factor in achieving customer satisfaction, reducing overall costs and creating the competitive position of various entities in the value chain |
| • Auxiliary activities are not structured so as to promote sustainable strategic management | • Enables the creation of networks of partner corporate groups belonging to a value chain that competes with other value chains |
| • Functions in a closed economic system without a further external ecosystem | • The philosophy of integration of the value chain is based on the idea of orientation to the market and the customer |

The consequence of this is the end of the traditional independent corporate group that withstands competition from other corporate groups individually, and a shift towards the formation of networks of partner corporate groups belonging to a value chain that competes with other value chains. The philosophy of integration of the value chain is based on the idea of orientation to the market and the customer [49,50]. Every activity in the value chain is directed towards the creation of value for the customer. Figure 1 shows the concept of a new business models based sustainable multiple customer value creation and specifies the areas to which it relates.

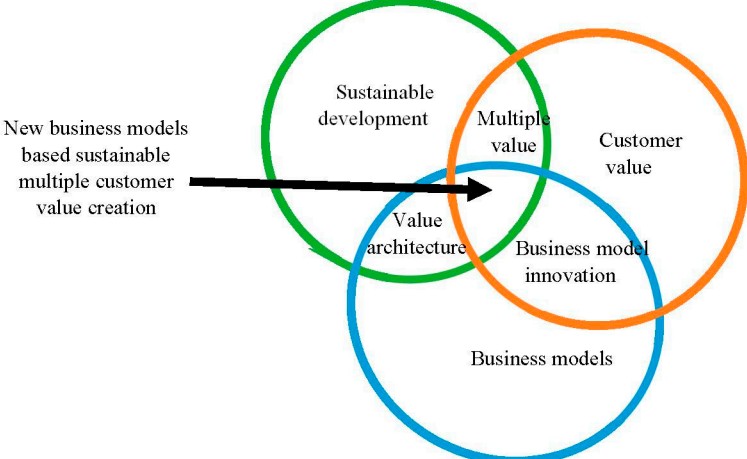

**Figure 1.** New business models focusing on the sustainable creation of multiple value for the customer from a broader perspective.

The formation of a value network. A value network makes it possible to gain an idea of the competences of corporate groups that may co-operate in the creation of value, while competing in communicating the value created [51]. According to Christensen and Rosenbloom [52], a value network is the context in which the corporate group competes and resolves the problems of customers. For this reason, it is appropriate to model a value network that incorporates stakeholders that engage the corporate group in the process of creating and providing value on the market. This involves stakeholders who create an expanded value chain from the stage before the drawing up of a product concept until after its distribution. Certain authors associate the concept of the value network with the modularity of products and systems that create the value architecture [53].

## 3. Methodology

To fulfil the article's objective, which is to describe business models currently used in terms of sustainable multiple customer creation in the Czech chemical industry, namely Section 20.1 according to CZ NACE, the following research questions have been formulated:

- Which business model elements promote sustainable multiple customer value creation?
- How to discern the elements which render the business model novel within the context of the industry?
- How to gauge the business model novelty?

The following paragraphs present the steps taken to answer the first two research questions from the perspective of the methodology used. The third research question will be answered in the results section. The methods, techniques, and tools used include the content analysis, semi-structured interviews, a questionnaire, benchmarking, and the Canvas business model as a visualization tool.

Description of the use of the method and techniques: The content analysis was used to compile elements of the business model. The semi-structured interviews were used to assess the representation and importance of individual elements for the examined industry from the basic set of elements

obtained. The questionnaire was used to identify the elements of business models of companies in the chemical industry and their customer orientation based on the sustainable creation of multiple value. Benchmarking was used to compare and measure business models of companies (using a weighted mean indicator of the creation of multiple value). The Canvas business model served as a visualisation tool.

### 3.1. Research Design–Five-Step Approach to Describe New Business Models Based on Creating Sustainable Multiple Customer Value

The set of relevant business model elements and novelty evaluation indicator were created through a five step process: (1) Extracting possibly relevant business model elements and their interpretations from the literature; (2) conducting content analysis to limit the initial set to a more manageable one; (3) semi-structured interviews with industry representatives to evaluate relevance; (4) questionnaire design, testing, and distribution; and (5) finding a measure of novelty.

### 3.2. Step One and Two: Literature Review and Content Analysis

Given the abstract nature of the business model construct, which is composed of elements that can have manifold interpretations and representations varying from practitioner to practitioner, the literature research was conducted in all three theoretical fields to provide a standardized set of relevant business model elements. The first result identified 199 elements (Table 2).

**Table 2.** Number of identified business model elements according to various offers.

| Number of Identified Elements | Author | Year |
|---|---|---|
| 11 | Hart, S. L. | 1995 |
| 17 | Kumar, S., Malegeant, P. | 2006 |
| 4 | Fowler, S. J., Hope, C. | 2007 |
| 5 | Dubigeon, O. | 2009 |
| 11 | Grandval, S., Ronteau, S. | 2011 |
| 7 | Chanal, V. | 2011 |
| 18 | Sempels, C., Hoffmann, J. | 2013 |
| 14 | Dauchy, D. | 2013 |
| 40 | Demil, B., Lecocq, X., Warnier, V. | 2013 |
| 6 | Rajagopal, R. | 2014 |
| 56 | Chen, Y.T., Chiu, M. | 2015 |
| 10 | Heitel, S., Kämpf-Dern, A., Pfnür, A. | 2015 |

This set was reduced through content analysis to tighten the spread of elements to those present in the industry. The content analysis included company websites, financial reports, annual reports, press releases, press articles, and other publicly accessible materials. The documents were scanned for specific mentions of business elements and causal factors which could indicate their presence. This reduced the initial set to a more manageable set of 53 elements relevant to the industry's context (Appendix A).

### 3.3. Step Three and Four: Semi-Structured Interview And Questionnaire Design

This set of 53 elements served as a foundation for semi-structured interviews with 9 industry representatives belonging to the companies present in the sample (production managers, commercial managers, and marketing managers). The representatives were asked to first confirm validity of the given elements, second evaluate on a Likert scale the significance to the industry's context and multiple value creation (1–not significant to 5–very significant). Table 3 (in Section 4.1) presents the most significant business model elements arranged according to the 9 fields of the business model canvas [54] to facilitate the comprehension and representation (Table 3). For the purpose of the research,

we modified 2 fields of the business model canvas, switching the fields "customer relationships" and "customer segments" for "stakeholder relationships" and "stakeholders".

**Table 3.** Reduced set of validated business model elements.

| Business Model Canvas Field | Business Model Element | Absolute Frequency | Frequency (%) | Minority Model |
| --- | --- | --- | --- | --- |
| Value proposition | Multiple product variants on offer | 32 | 84.21 | |
| | Alternatives to products on offer | 17 | 44.74 | X |
| | Environmentally friendly products | 24 | 63.16 | |
| | Related products on offer | 16 | 42.11 | X |
| | Individual planning | 23 | 60.53 | |
| | Limiting use of dangerous substances in production | 23 | 60.53 | |
| Key partners | Suppliers of support services | 22 | 57.89 | |
| | Local suppliers | 16 | 42.11 | X |
| | Emphasis on industrial safety | 27 | 71.05 | |
| | Cooperation with public and non-profit organisations | 11 | 28.95 | X |
| Key activities | B2B resource sharing | 4 | 10.53 | X |
| | Commercial support for B2B customers | 21 | 55.26 | |
| | Using sustainable feedstocks | 18 | 47.37 | X |
| | Using energy saving equipment | 21 | 55.26 | |
| Key resources | Centralised waste treatment | 25 | 65.79 | |
| | Take-back agreement | 10 | 26.32 | X |
| Stakeholder relations | Consulting | 27 | 71.05 | |
| | Information and report sharing | 23 | 60.53 | X |
| | Sharing experience with customers and suppliers | 16 | 42.11 | X |
| | Research cooperation | 18 | 47.37 | X |
| | Provision of internships | 20 | 52.63 | |
| Channels | Online platform | 13 | 34.21 | X |
| Cost structure | Waste as an energy resource | 15 | 39.47 | X |
| | Centralised waste treatment | 25 | 65.79 | |
| | Waste recycling | 23 | 60.53 | |
| | Financial support and sponsorships | 24 | 63.16 | |
| | Education fund | 5 | 13.16 | X |
| Revenue streams | Rental of production facilities and equipment | 7 | 18.42 | X |
| | Specialised services (R&D) | 21 | 55.26 | |
| | Complete product-service solutions | 3 | 7.89 | X |
| | Maintenance | 24 | 63.16 | |
| | By-products available as energy source or resource | 11 | 28.95 | X |

This secondary set was reduced based on indicated validity, and elements with less than 50% validity were excluded. Thus, based on this final set of 32 elements, a questionnaire was developed and distributed from October 2017 to February 2018. Due to the nature of business model elements (being present or non-present), the questionnaire was designed as a series of 32 closed questions. The questionnaire consisted of two major parts, questions concerning business model elements and questions concerning customers as part of their stakeholders. The questions in the first part of the questionnaire concerned the value offer–comprising 32 elements obtained by the content analysis (verified by structured interviews). They were divided into the following areas, which were specified by individual questions: Types of services provided (4 questions), Diversity of offer (4 questions), Availability of products and services (7 questions), Environment (8 questions), and Social environment (9 questions). Also, other relevant stakeholders were attributed to some questions (e.g., the stakeholder "Universities" was attributed to element "Provision of internships"). This is later represented in Figures 2 and 3.

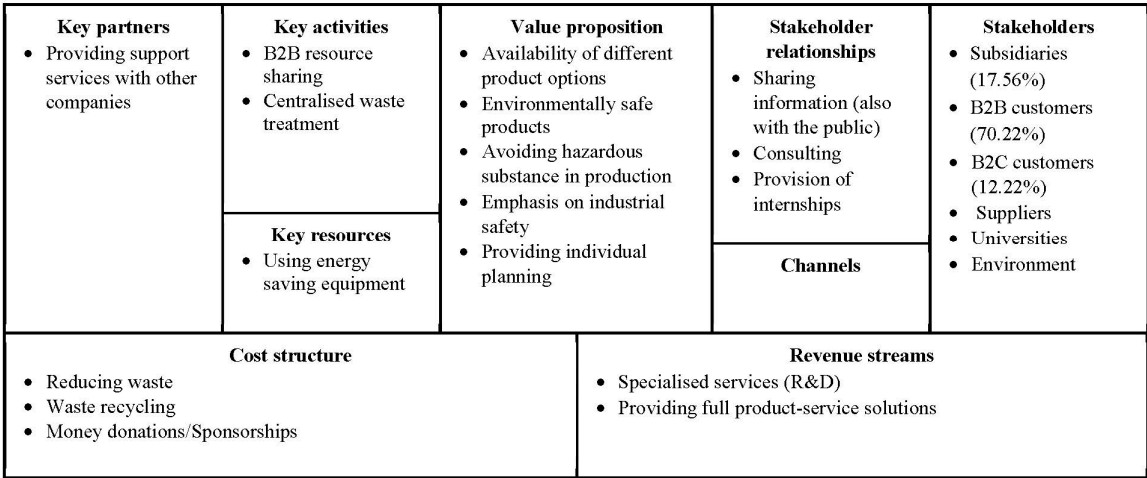

| Key partners | Key activities | Value proposition | Stakeholder relationships | Stakeholders |
|---|---|---|---|---|
| • Providing support services with other companies | • B2B resource sharing<br>• Centralised waste treatment | • Availability of different product options<br>• Environmentally safe products<br>• Avoiding hazardous substance in production<br>• Emphasis on industrial safety<br>• Providing individual planning | • Sharing information (also with the public)<br>• Consulting<br>• Provision of internships | • Subsidiaries (17.56%)<br>• B2B customers (70.22%)<br>• B2C customers (12.22%)<br>• Suppliers<br>• Universities<br>• Environment |
| | **Key resources**<br>• Using energy saving equipment | | **Channels** | |

| Cost structure | Revenue streams |
|---|---|
| • Reducing waste<br>• Waste recycling<br>• Money donations/Sponsorships | • Specialised services (R&D)<br>• Providing full product-service solutions |

**Figure 2.** Majority business model construct.

| Key partners | Key activities | Value proposition | Stakeholder relationships | Stakeholders |
|---|---|---|---|---|
| • Procurement from local suppliers<br>• Cooperation with public/non-profit organisations | • B2B resource sharing<br>• Providing take-back agreement | • Availability of product alternatives<br>• Availability of related products | • Assisting / participating in academic research<br>• Sharing experience (with customers and suppliers) | • Subsidiaries (17.56%)<br>• B2B customers (70.22%)<br>• B2C customers (12.22%)<br>• Suppliers<br>• Local government<br>• Public/non-profit organisations<br>• Environment |
| | **Key resources**<br>• Sustainable feedstocks | | **Channels**<br>• Online platform for reservation | |

| Cost structure | Revenue streams |
|---|---|
| • Waste to energy<br>• Establishing an education fund | • Provision of maintenance<br>• B2B Leasing<br>• Sale of waste materials e as a resource or energy |

**Figure 3.** Minority business model construct.

The questionnaire was mainly aimed at production managers, marketing managers, or commercial managers. Prior to the distribution, the questionnaire was tested using semi-structured interviews with different two practitioners to check for comprehension. The participants were also asked about their comprehension and compared to set definitions of individual elements. The definitions were not provided to the practitioners but served to check the comprehension.

The sample comprised medium-sized and large enterprises falling into the CZ NACE group 20.1 manufacture of basic chemicals, fertilizers, and plastics. The companies were found using the Amadeus database and sorted according the following criteria: (1) Active status; (2) having the headquarters or a branch in the Czech Republic; (3) number of employees higher than 75 employees; (4) belonging to the group 20.1 manufacture of basic chemicals, fertilizers, and plastics.

The first results indicated a population of 49 companies in total, using the Amadeus database. Following a review, double entries and false entries were eliminated, thus leaving a population of 42 companies. After contacting individual companies when distributing the questionnaire, commercial outlets and wholesalers were almost all eliminated from the population, thus providing a final number of 38 companies. These 38 companies consist of 25 large and 13 medium-sized companies. After contacting all 38 companies represented in the sample, 22 questionnaires were received, representing a 57.89% response rate.

*3.4. Step Five: Finding a Measure of Novelty*

The further theoretical research focused on quantifying business model novelty. The first attempts were aimed and associating the business model with its performance as a proxy to a measure of novelty. This was under the supposition that by introducing a New business model element into the business model will affect the company's performance. Namely the sustainable value-added approach by Figge and Hahn [55] seemed promising due to its incorporation of all three aspects of multiple value creation based on the triple bottom line concept [56] and the industry's ability to precisely gauge its environmental impacts. This approach proved to be problematic, due to the industry's sensitive public perception and sensitive nature of required data (CO2, Sox, greenhouse gas emissions, etc.). Thus, a separate indicator was developed for gauging business model novelty. The business model novelty indicator (1) is based on the sum of weights for each element present in the company's business model (Appendix A). The data later served to benchmark the novelty of business models present in the industry. The benchmark was calculated as a weighted average of corresponding BMNI values.

$$BMNI = \sum_{i=0}^{n} EI_i * f_i \qquad (1)$$

where: $n$–number of business model elements; $EI_i$–significance of a business model element; $f_i$–presence of a business model element can only achieve the value of 0 or 1.

## 4. Results

The following section presents the answers to the three research questions specified above, therefore, the results of the five-step approach described in the previous section. The results include the experts' agreed-on relevant business model elements promoting the sustainable multiple customer value creation within the context of the industry, a majority business model construct, a minority business model construct [1], and a benchmark of the currently used business models. To give greater context, derived also from theory, it is necessary state the relevant stakeholder for which the industry creates, delivers, and captures value. The first part of the questionnaire (Section 3.3) served to this end. According to the questionnaire results, companies in the industry focus on subsidiaries (17.56%), B2B customers (70.22%), and B2C customers (12.22%) as customers. Other stakeholders were derived from marking specific answers in the questionnaire.

*4.1. Which Business Model Elements Promote Sustainable Multiple Customer Value Creation?*

Table 3 presents the reduced set of validated business model elements arranged according to the nine fields of the business model canvas along with their frequency of presence in the industry and their attribution to the minority model. The data presented in the table served as a foundation to describe the industry's business model and new business models and answer the following two research questions: Which business model elements promote sustainable multiple customer value creation? How to discern the elements which render the business model novel within the industry's context?

*4.2. How to Discern the Elements Which Render the Business Model Novel within the Industry's Context?*

To discern new business models, there must be a distinction to what is novel and what is common use in the industry's context. Therefore, the reduced set in Table 2 need to be divided into two subsets. This means a subset representing the most commonly used elements subset and incorporating the novel elements. This division is based on the assumption that new business model elements will be the ones less frequently represented in the entire set of elements. In this case, the division is therefore based on an element's frequency (Table 3), with the subsets being represented as the majority business model and the minority business model. If the frequency of an element is higher than 51%, it is attributed to the majority business model and vice versa.

The majority business model design (Figure 2) indicates the most prevalent business model elements. For this reason, it indicates a business model employed by the majority of companies in the industry and which would be required to follow if a new rival penetrated the market. From the perspective of the narrative approach [57], the majority business model represents the general narrative of the industry.

The minority business model design (Figure 3) represents a range of differentiating elements, which are geared towards satisfying the environmental and social expectations. The minority model design displays the orientation towards a larger number of stakeholders, new income streams, as well as opportunities to reduce opportunity costs and drive eco-efficiency.

*4.3. How to Gauge Business Model Novelty?*

Based on the two business model designs (Figures 2 and 3), there could be an assumption formulated that once a minority business model element is introduced into a business, then it can be deemed novel within the industry's context. However, research showed that the business models present in the industry are made of a mix of business model elements native to both constructs. Therefore, the business model novelty indicator (1) was designed and employed. The indicator resolves the problem with the business model being constructed from elements from both constructs. Moreover, if a business employs several minority business model elements, the indicator increases further showing the difference in novelty among business models.

Based on the value calculated using BMNI a benchmark was calculated at a value of 561.12. Figure 4 presents the order of companies according to the benchmark value. Those which have a higher value than the benchmark can be deemed as companies utilizing a new business model within the industry's context. Those with value close to the benchmark value either are reaching the limit of their extant business models or have started the process of changing their business model.

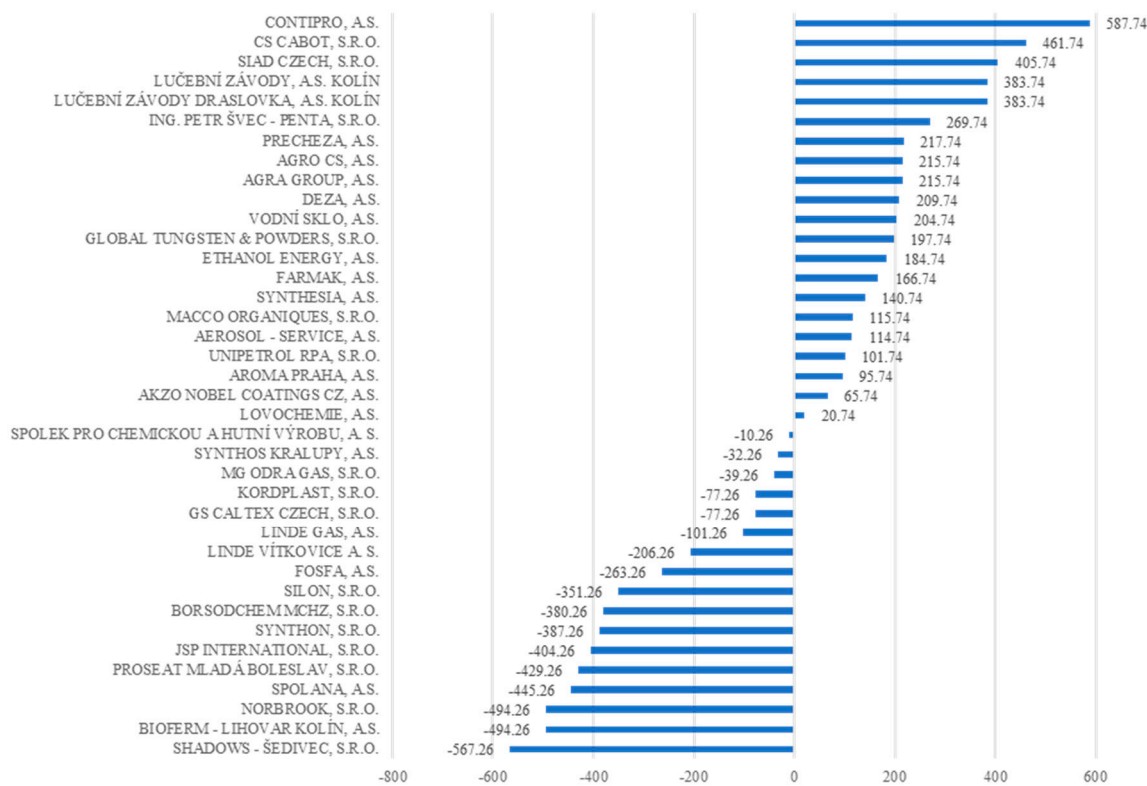

**Figure 4.** Benchmark of business model present in the Czech chemical industry based on the Business Model Novelty Indicator.

## 5. Discussion and Conclusions

The aim of our research project was to respond to the current challenges of sustainable development in the form of developing new business models. In the paper, we presented how we grasped this challenge in the case of the selected CZ NACE group of the Czech chemical industry.

The presented research is an attempt to formulate a methodology from industry-wide business model research. Although the cited authors in the article whose approaches we used or considered while designing the research provide possible avenues for conducting industry-wide research [33,41,58–69], the available body of knowledge remains dispersed. For further research of this sort we can provide the industry's prevalent narrative and limits of the utilized methodology.

### 5.1. The Industry's Prevalent Narrative

From the perspective of the multiple customer value creation, the prevailing narrative of the industry consists in the fact that companies offer different product options and individual planning solutions for their customers and at the same time, they emphasise the environmental safety of their products, industrial safety, as well as avoiding hazardous substances in their output. They achieve this by means of offering support services to their customers in cooperation with other companies, B2B resource sharing, centralised waste treatment, and using energy-saving equipment. Relationships with individual stakeholders are upheld through sharing information about the company's operations, consultancy, and offering internships to university students. The cost structure comprises promoting cost cutting originating from waste reduction and recycling it, as well as addressing the costs of a fund for donations and sponsorships. Revenue streams include providing specialised services such as research and development and complete-service product solutions. This narrative is adapted using elements functioning as differentiating parts (Figure 4). Thus, the adoption of some business model elements promoting sustainable multiple value creation is widespread, as presented by the majority business model construct. However, mainly medium-sized companies have reached the limit of the majority business model and lead of creation of new business models within the industry's context.

### 5.2. Limits of Utilized Research

The research does not indicate the order in which individual business model elements are implemented. Each individual business model is composed of a unique combination of elements belonging to the majority or minority business model. Thus, stating that the threshold of novelty is based on the usage of all majority business model elements is flawed. On the other hand, it would be rather forgiving to claim that once a company adopts and sustains an element from the minority business model that it employs a new business model, once the elements impact has been demonstrated by a change in the performance. The minority model shows the least common characteristics of companies. They are emerging, little used, or marginal elements in business. The minority business model has the characteristics that are least represented in the particular industry. It may carry rare yet promising sustainable innovations. The model creates multiple value using less-common activities. In general, the elements of the minority business model are still under development and have not been standardised and generally used or they are typical only for some companies.

The research does not show an apparent validation of the connection to greater performance. By selecting companies that employ at least one minority business model element and putting those into clusters showed four significant groups consisting of mainly medium-sized companies (Appendix B). The comparison of available financial indicators, namely the net profit, added value, and return on equity, revenues, profit margin, cash flow, and return on assets, showed no proof among clusters or among companies within a cluster.

The narrative based approach provides only a static image of the business model's current state [57]. This goes hand in hand with the last point where the adoption and sustaining of a business model element will be visible in a company's performance indicators in the years to come. Therefore,

a dynamic approach would maybe be more suitable, but this still would provide its own issues to resolve, e.g., distinguishing whether the change in the studied company's performance is caused by adopting a new business model element or whether the change was caused by a market shift.

Once a company reaches the limits of the industry's standard business model (the majority business model) it searches to employ new business model elements to provide greater value for its customers. Gauging this novelty on an industry wide scale proves to be problematic, due to the static nature of the narrative approach of business models. A dynamic approach may offset the presented research limits.

The practical benefits and implications of the research lie in identifying the most important elements of the new business models, based on which the existing majority model of the industry has been identified, as well as minority models of the surveyed companies, which are further decomposed on the basis of the relevant elements.

This allows companies to create new ways of doing business, while other stakeholders are able to better understand the whole process of creating multiple value for the customer based on the visualisation of the situation (in the form of CANVAS) in the companies participating in the research.

**Author Contributions:** I.Š. and P.K. have contributed to all parts of the manuscript. This includes conceptualization, research designing, analysis, investigation, resources, writing-original draft preparation, review & editing. All authors have read and agreed to the published version of the manuscript.

**Funding:** This research received no external funding.

**Acknowledgments:** This paper received support within the research project entitled. The Concept of Business Management and Development in an Environment of Multidisciplinary Value Creation Networks, which is founded by Specific Research of FBM BUT, grant number FP-S 20-6355.

**Conflicts of Interest:** The authors declare no conflict of interest.

## Appendix A

**Table A1.** Elements of the initial business model (after content analysis).

| Category | Element Designation |
|---|---|
| **Cooperating Companies** | Use of marketing intermediaries<br>Self-realization in the company itself<br>Cooperation with other partners |
| **Services** | Product lease<br>B2B product lease<br>Specialised services (R&D)<br>Lease of production capacities, machinery and equipment<br>B2B recourse sharing<br>B2B customer support<br>Consultancy<br>Complete service<br>Retail network<br>Additional services<br>Suppliers of support services<br>Sales on a deduction basis |

**Table A1.** *Cont.*

| Category | Element Designation |
|---|---|
| **Products** | Offer of multiple product variants |
| | Modified products |
| | Secondary materials as a source of energy and raw materials |
| | Product alternatives |
| | Environmentally safe products |
| | Offer of a single product model |
| | Finished product |
| | Work-in-progress |
| | On-line platform |
| | Maintenance |
| | Availability of similar products |
| | Availability of multiple product variants |
| | Offer of the additional assortment |
| | Individual planning |
| | Customer support itself |
| **Environmental Aspect** | Reducing the use of hazardous substances in production |
| | Packaging made of environmental-friendly materials |
| | Use of sustainable sources of raw materials (feedstocks) |
| | Use of energy saving equipment |
| | Waste as a source of energy |
| | Centralised waste collection |
| | Water recycling |
| | Water resources management |
| | Reduction of waste in production |
| | Waste recycling |
| | Central waste treatment |
| | Minimisation of disposable products |
| | Take-back after consumption |
| | Greenhouse gas emissions management |
| **Social Aspect** | Local suppliers of raw materials |
| | Use of high-quality materials |
| | Industrial safety |
| | Public and non-profit organisations |
| | Information and reporting sharing |
| | Sharing experience with customers and suppliers |
| | Research cooperation |
| | Financial donations/sponsorship |
| | Placements |
| | Training fund |

## Appendix B

Company names are not provided in order to maintain confidentiality.

In most cases, the examined sample contains medium-sized enterprises, which have their business model better adapted to creating the multiple value for the customer. From the mentioned clusters it is the cluster No. 4 that has the best adapted models for creating the multiple value for the customer according to the specific parameters.

**Table A2.** Clusters of surveyed companies (according to the reference number).

| | Reference Number | Sum of Elements in the Business Model | Number of Elements from the Minority Model | Share of Elements from the Minority Model | MVI | Average MVI | Average Share of Elements from the Minority Model |
|---|---|---|---|---|---|---|---|
| 1 | 21 | 15 | 3 | 0.20 | 588 | | |
| | 12 | 20 | 5 | 0.25 | 734 | 692.25 | 0.23 |
| | 14 | 20 | 5 | 0.25 | 765 | | |
| | 1 | 18 | 4 | 0.22 | 682 | | |
| 2 | 27 | 21 | 7 | 0.33 | 785 | | |
| | 10 | 21 | 9 | 0.43 | 777 | | |
| | 25 | 14 | 5 | 0.36 | 528 | 684 | 0.32 |
| | 38 | 21 | 7 | 0.33 | 772 | | |
| | 15 | 13 | 3 | 0.23 | 490 | | |
| | 11 | 20 | 5 | 0.25 | 752 | | |
| 3 | 24 | 19 | 5 | 0.26 | 683 | | |
| | 5 | 18 | 6 | 0.33 | 663 | | |
| | 4 | 17 | 4 | 0.24 | 663 | 715 | 0.28 |
| | 2 | 21 | 6 | 0.29 | 783 | | |
| | 3 | 21 | 6 | 0.29 | 783 | | |
| 4 | 23 | 26 | 11 | 0.42 | 951 | | |
| | 22 | 26 | 11 | 0.42 | 951 | | |
| | 16 | 23 | 9 | 0.39 | 837 | 982.67 | 0.42 |
| | 9 | 28 | 12 | 0.43 | 1029 | | |
| | 8 | 32 | 15 | 0.47 | 1155 | | |
| | 30 | 27 | 11 | 0.41 | 973 | | |

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
