# Peer review of "New Business Models Based on Multiple Value Creation for the Customer: A Case Study in the Chemical Industry"

_sustainability, doi:10.3390/su12093932_

Round 1

Reviewer 1 Report

See file attached.

Author Response

Dear Editor and Reviewers,

Thank you for all your comments and effort to help us improve our paper. We hope, that we have incorporated into the text all your recomandations to do the text more understanable and readable. In the text we used "track changes" and put also the comments what and why we do it according to revievers (R1, R2). We wrote in the comments R1 or R2 or both (when both rewievers requested this change in the some way)

Generally we would like to give comments, which is not  regarding special lines in the text:

The corrected paper  subsequently underwent by english proofreading.

We did right order of tables adn renumberation. Instead additional tables we added 2 Appendixes (A and B), App.A shows the List of elements after the content analysis and app.B shows the results - Clusters of surveyed companies. After your reccomendations and changes in the text we feel that it could be all more understandable and the main reason is, that we dont have any appropriate evidence of semi-structured interviewes (interwievs were recordeded on the place and  used for correction and validation of elements for the researched sector). We described the methods, tools and techniques, which were  used in the research. We added some information about the used questionnaire. But it is not possible to put in the paper all the details. We hope, that now after changes, will be our text more uderstandable and clear, also in methodological issues.

In section 5 Discussion and conclusions- we added clear implications for stakeholders.

We also incoporated another recomended sources in the list of references and quoted into the text.

Thank you one more for any further comments

Kind regards

Authors

Reviewer 2 Report

The paper addresses a very interesting and needed topic, that is, the development of new business models within the perspective of sustainability taking in synchronisation a number of values (economic, social and environmental) and being anchored together with all aspects of the multiple value. The authors do an interesting debate, a well-balanced theoretical background introduction, prompting the reader to a very actual topic, namely the concerns to assure the attainment of the goal of economic efficiency in line with social justice and protection of the environment. Moreover, the option for the chemic industry as the empirical approach sounds quite interesting and opportune.

Albeit, the manuscript sounds interesting, well written and well-structured some small improvements can be done, in order to be published, namely:

  • The paper still needs some English proof editing, as well as some careful attention to sections such as for example, in: in the abstract please check the writing ‘The research is stems from the theoretical background and anticipates that the business model development dynamics drives companies to consider the reasons and conditions of their very existence’; Table 1. A differing approach to the value chain from the viewpoint of creation of economic, environmental and social value – please check the titles in the table; after fig. 1 please check the beginning of the last paragraph at the end of section 2.6.; at the end of section 3.2 its mentioned an appendix A, however I cannot find it; please review first sentence of section 5.1; rewrite the sentence ‘On the other hand, stating that once a company adopts and sustains an element from the minority business model than its employs a new business model may be a more forgiving statement once the elements impact is proved by a change in performance’ in section 5.2;
  • Concerning the empirical part the following suggestions must be edited:
  • When presenting the empirical methods followed, the authors should base it on previous approaches and methodologies, justifying with prior works;
  • The authors made semi-structured interviews with 9 industry representatives belonging to the companies present in the sample (production managers, commercial managers, and marketing managers) – please describe and insert a table characterizing these interviewees;
  • The questionnaire was mainly aimed at production managers, marketing managers, or commercial managers. 38 companies made part of the sample, 22 answers - please describe and insert a table characterizing these interviewees;
  • It is needed to provide a more detailed description of the questionnaire used;
  • Section 3.2 must be more complete, denoting clearly the steps comprised, since identification, to screening, to eligible contributions and the included. Probably more information, graphical one could improve the readers’ understanding of the process and extracted information – possibly using Vosviewer;
  • In the section 5 Discussion and conclusions, I would like to see also the implications for the stakeholders involved, companies, managers, consumers, suppliers, policy makers, etc..
  • I see the authors have added one citation from the present journal which improve the quality of the research, although please use the following to cite: Sustainable Business Models, Adam JabÅ‚oÅ„ski (Ed.) Published: January 2019. by the authors; CC BY-NC-ND license, https://doi.org/10.3390/books978-3-03897-561-8; Leitão, J and Pereira, D. (2016). Absorptive Capacity, Coopetition and Generation of Product Innovation: Contrasting Italian and Portuguese Manufacturing Firms, International Journal of Technology Management 71-1/2 (2016), DOI: 10.1504/IJTM.2016.077979; Leitão, J. (2018). Open Innovation Business Modeling: Gamification and Design Thinking Applications. DOI: 10.1007/978-3-319-91282-0, Publisher: Springer International Publishing, ISBN: 978-3-319-91281-3.

Considering the previous comments, and although I recognize potential to the ongoing research, I recommend the referred revisions of the manuscript.

Author Response

Dear Editor and Reviewers,

Thank you for all your comments and effort to help us improve our paper. We hope, that we have incorporated into the text all your recomandations to do the text more understanable and readable. In the text we used "track changes" and put also the comments what and why we do it according to revievers (R1, R2). We wrote in the comments R1 or R2 or both (when both rewievers requested this change in the some way)

Generally we would like to give comments, which is not  regarding special lines in the text:

The corrected paper  subsequently underwent by english proofreading.

We did right order of tables adn renumberation. Instead additional tables we added 2 Appendixes (A and B), App.A shows the List of elements after the content analysis and app.B shows the results - Clusters of surveyed companies. After your reccomendations and changes in the text we feel that it could be all more understandable and the main reason is, that we dont have any appropriate evidence of semi-structured interviewes (interwievs were recordeded on the place and  used for correction and validation of elements for the researched sector). We described the methods, tools and techniques, which were  used in the research. We added some information about the used questionnaire. But it is not possible to put in the paper all the details. We hope, that now after changes, will be our text more uderstandable and clear, also in methodological issues.

In section 5 Discussion and conclusions- we added clear implications for stakeholders.

We also incoporated another recomended sources in the list of references and quoted into the text.

Thank you one more for any further comments

Kind regards

Authors

This manuscript is a resubmission of an earlier submission. The following is a list of the peer review reports and author responses from that submission.